# Increasing Muscle Mass in Elders through Diet and Exercise: A Literature Review of Recent RCTs

**DOI:** 10.3390/foods12061218

**Published:** 2023-03-13

**Authors:** Gavriela Voulgaridou, Sophia D. Papadopoulou, Maria Spanoudaki, Fovi S. Kondyli, Ioanna Alexandropoulou, Stella Michailidou, Paul Zarogoulidis, Dimitris Matthaios, Dimitrios Giannakidis, Maria Romanidou, Sousana K. Papadopoulou

**Affiliations:** 1Department of Nutritional Sciences and Dietetics, School of Health Sciences, International Hellenic University, 57400 Thessaloniki, Greece; 2Laboratory of Evaluation of Human Biological Performance, Department of Physical Education and Sport Science, Aristotle University of Thessaloniki, 57001 Thessaloniki, Greece; 3Department of Cardiology, Barts Heart Centre, Barts Health NHS Trust, London EC1A 7BE, UK; 4Pulmonary Department, General Clinic Euromedica Private Hospital, 54645 Thessaloniki, Greece; 53rd Surgery Department, AHEPA University Hospital, Aristotle University of Thessaloniki, 54636 Thessaloniki, Greece; 6Oncology Department, General Hospital of Rhodes, 85100 Rhodes, Greece; 71st Department of Surgery, Attica General Hospital “Sismanogleio—Amalia Fleming”, 57889 Athens, Greece; 8Adult Eating Disorders Service, Essex Partnership University NHS Foundation Trust, Wickford SS11 7XX, UK

**Keywords:** muscle mass, sarcopenia, older adults, exercise, resistance exercise, combination of exercise, diet, protein, amino acids

## Abstract

This study aimed to review the current evidence on the independent and combined effects of diet and exercise and their impact on skeletal muscle mass in the elderly population. Skeletal muscle makes up approximately 40% of total body weight and is essential for performing daily activities. The combination of exercise and diet is known to be a potent anabolic stimulus through stimulation of muscle protein synthesis from amino acids. Aging is strongly associated with a generalized deterioration of physiological function, including a progressive reduction in skeletal muscle mass and strength, which in turn leads to a gradual functional impairment and an increased rate of disability resulting in falls, frailty, or even death. The term sarcopenia, which is an age-related syndrome, is primarily used to describe the gradual and generalized loss of skeletal muscle mass (mainly in type II muscle fibers) and function. Multimodal training is emerging as a popular training method that combines a wide range of physical dimensions. On the other hand, nutrition and especially protein intake provide amino acids, which are essential for muscle protein synthesis. According to ESPEN, protein intake in older people should be at least 1 g/kgbw/day. Essential amino acids, such as leucine, arginine, cysteine, and glutamine, are of particular importance for the regulation of muscle protein synthesis. For instance, a leucine intake of 3 g administered alongside each main meal has been suggested to prevent muscle loss in the elderly. In addition, studies have shown that vitamin D and other micronutrients can have a protective role and may modulate muscle growth; nevertheless, further research is needed to validate these claims. Resistance-based exercise combined with a higher intake of dietary protein, amino acids, and/or vitamin D are currently recognized as the most effective interventions to promote skeletal muscle growth. However, the results are quite controversial and contradictory, which could be explained by the high heterogeneity among studies. It is therefore necessary to further assess the impact of each individual exercise and nutritional approach, particularly protein and amino acids, on human muscle turnover so that more efficient strategies can be implemented for the augmentation of muscle mass in the elderly.

## 1. Introduction

Aging is associated with a generalized deterioration of physiological function, accompanied by a progressive decline in skeletal muscle mass, and strength, gradually leading to functional impairment, increased disability, and dependency [1]. The risks of falls, loss of independence, and premature death have been progressively rising in the last few decades among the elderly population [2,3]. Moreover, these burdens are linked to a higher economic burden on the healthcare system [4]. In addition to the loss of muscle mass due to aging, low muscle mass is observed as a result of diseases such as cancer, chronic obstructive pulmonary disease, heart failure, and renal failure and is a prognostic indicator in a multitude of diseases [5,6,7,8,9,10]. The prevalence of sarcopenia varies according to the population or according to measurement tools [11]. In a meta-analysis by Papadopoulou et al. [11], the prevalence of sarcopenia ranged between 9–11% in women and men, respectively, in community-dwelling older adults. The prevalence of sarcopenia increases dramatically in nursing-home residents and hospitalized individuals, approaching 51% in men and 31% in women, and about 23% in both men and women, respectively [11].

Maintaining muscle mass in populations due to pathophysiological and/or pathoanatomical conditions is difficult without appropriate intervention. The combination of exercise and dietary supplementation with a specific amino acid mixture appears to have similar histopathological, biochemical, and functional changes in elderly patients [12]. A study has gathered evidence that the intake of essential amino acids can improve the exercise capacity of people with chronic heart failure or chronic obstructive pulmonary disease [12]. This improvement is due to the fact that they increase muscle mass and function, improve their aerobic metabolism, and increase their sensitivity to insulin [12]. Their ability to do the above appears to derive from the mechanisms they activate in muscle tissue, namely mitochondrial generation and myofibril growth, as well as glucose and cardiac muscle control [12]. In contrast with healthy adult exercisers and sportsmen, who maintain or even increase muscle mass through exercise and diet following current recommendations, elder people with morbidities and co-morbidities require particular attention, as a pathophysiological decline in muscle mass leads to other health problems [13], perpetuating a vicious cycle. 

Muscles play an essential role in performing daily activities. In humans, skeletal muscles comprise approximately 40% of total body weight and contain 50–75% of all body proteins [14]. Maintaining or increasing muscle mass is a key component for improving daily activities as well as sports performance in activities of daily living and sports [15]. On the other hand, accelerated loss of muscle mass and function is associated with increased adverse outcomes, including falls, functional decline, frailty, and mortality [16].

Imaging methods such as X-ray or computed tomography (CT) are the gold standard for measuring muscle mass [17]. However, these methods are not used in daily clinical practice due to several limitations, such as their high cost. Most commonly, dual-energy X-ray absorptiometry (DXA) is used to estimate appendicular skeletal muscle mass/appendicular soft tissue (ASMM) or lean body mass (fat-free mass, total lean soft tissue plus bone mass, or body weight minus fat mass) as a proxy for muscle mass [18]. A simpler and even non-invasive technique of measuring muscle mass is bioelectrical impedance analysis (BIA), which is used for the estimation of fat-free mass (FFM), ASMM, or even total body skeletal muscle mass (SMM) through predictive equations [19]. According to the literature, the range of published cut-off values for ASMM by DXA is 5.86–7.40 kg/m^2^ in men and 4.42–5.67 kg/m^2^ in women, whereas cut-offs through the BIA analysis method range from 6.75–7.40 kg/m^2^ in men and 5.07–5.80 kg/m^2^ in women [19]. In healthy populations, cut-off values based on CT or magnetic resonance imaging (MRI) methods ranged between 36.54 and 45.40 cm^2^/m^2^ in men and 30.21 and 36.05 cm^2^/m^2^ in women [18].

The skeletal muscle groups of the human body consist of muscle fiber bundles. At the level of the whole muscle, its size is mainly determined by the number and size of individual muscle fibers [14]. The two most abundant myofilaments (proteins) are actin and myosin, which together comprise approximately 70–80% of the total protein content of a single fiber [14]. Skeletal muscle fibers are grouped into two categories: type I, called slow twitch, and type II, called fast twitch [20]. Type II muscle fibers can be grouped in terms of how myosin chains are expressed into three classes: IIa, IIx, and IIb, which are not present in our species [14]. A single muscle fiber may simultaneously express more than one type of myosin heavy chain; for example, types I and IIa or IIa and IIx together [14].

Altered expression in the myosin heavy chain may be the result of mRNA transcriptional activity in different nuclear regions [20]. These hybrid fibers appear to have two main roles: i. they function as mediators during fiber-type transitions associated with skeletal muscle development, exercise adaptation, and aging; and ii. they provide a functional continuum of fiber phenotypes, as they possess physiological properties that are intermediate to those of pure fiber types [21]. 

Muscle mass size is determined by coordinated changes in muscle protein synthesis and breakdown [22]. The combination of exercise and diet is a potent anabolic stimulus through the stimulation of muscle protein synthesis by amino acids [23]. Increased muscle mass is due to the accumulation of small amounts of protein in response to each bout of exercise combined with nutrient intake [23,24]. The effect of dietary supplements on muscle metabolism and muscle loss and regain has also been investigated and will be discussed below. 

Metabolism, i.e., the set of reactions of the organism, consists of metabolic pathways in which molecules of chemical compounds are converted into others through processes catalyzed by enzymes and facilitated by other chemical compounds, such as cofactors, coenzymes, and metabolic intermediates [25]. Nutrients are molecules that participate in and influence the reactions of metabolism in general, therefore also muscle metabolism. Nutrients can be obtained through the diet, but in cases of deficiency, especially in the elderly, they can also be taken in the form of supplements.

Nutrition plays a key role in the prevention and treatment of sarcopenia. Energy intake decreases by almost 25% between 40 and 70 years of age, leading to inadequate intake of macro- and micronutrients [26]. Nutritional intake is often compromised in older people, and the risk of malnutrition is increased. According to ESPEN (the European Society for Clinical Nutrition and Metabolism), the presence of either a high unintentional loss of body mass (>5% at six months or >10% beyond six months) or a markedly reduced body mass index (BMI) (<20 kg/m^2^) or muscle mass should be considered a serious indication of malnutrition that requires clarification of the underlying causes [27]. 

The total energy expenditure in older individuals is 24–36 kcal/kg body weight (BW) [27]. The basic energy requirements are influenced by lean body mass, gender, and nutritional status. Total energy expenditure is higher for men than for women and increases with decreasing BMI [27]. Maintaining or increasing body weight and muscle mass are central goals of nutritional support. Adequate protein intake of high biological value provides all essential amino acids, whereas both vitamin D and B12 have been studied for the prevention and treatment of sarcopenia [26].

There is a growing body of literature demonstrating how nutritional supplements affect muscle metabolism and muscle mass. Vaughan et al. [28] have shown, in in vitro experiments in human rhabdomyosarcoma cells, that mitochondrial biosynthesis increases within one day when OxyElite Pro and Cellucore HD dietary supplements are administered. These contain, among other ingredients, germanium and caffeine, respectively. The oxidative and glycolytic capacities of the cells also increased [28]. The effect of creatine on muscle metabolism is the subject of various research studies. It has been suggested that it acts as a positive stimulus for protein synthesis pathways, either as a cell stressor or by targeting the mTOR pathway [29]. It even affects the synthesis of muscle fibers, altering the secretion of myokines or regulatory factors that increase the mitotic divisions of cells that will differentiate into myofibrils [29]. L-carnitine supplementation seems to increase muscle mass, but due to increased levels of a potentially atherogenic compound (fasting plasma trimethylamine-N-oxide, TMAO), further research is required before long-term supplementation is recommended [30]. At the amino acid level, in vitro and in vivo animal studies have shown that the amino acids serine and glycine are required for muscle regeneration [31]. In addition, serine levels have been found to decrease in older individuals, so it is hypothesized that serine supplementation may benefit muscle regeneration [31].

Regarding vitamin supplements, meta-analyses show that supplementation with vitamin D has a catalytic effect on muscle strength but not on muscle mass in the elderly and especially in those with initially low serum levels of this vitamin [32]. In contrast, more recent studies do not show that muscle in the elderly population is affected by the intake of vitamin D supplements [33]. More and better-designed studies are needed to accurately determine the effects on the human population. Additionally, the various forms of vitamin K appear to affect energy metabolism in skeletal muscle by increasing ATP production and maintaining the mitochondrial number and satellite cell proliferation in muscle [34].

Exercise is the main method for increasing muscle mass. It is also well-documented that training can promote muscle protein synthesis and activate signaling pathways that regulate muscle fiber metabolism and function [35]. Exercise can be a method to treat muscle atrophy in many of these conditions. Mitochondria play an important role in skeletal muscle homeostasis and bioenergy metabolism. Mitochondria are sensitive to contractile signals, and therefore, exercise can improve mitochondrial function and promote biosynthesis, which ultimately maintains the healthy state of cells and the whole body [36]. Exercise can activate the signaling pathway that stimulates skeletal muscle fiber metabolism and promotes contraction and physiological muscle function [36]. In particular, exercise, regardless of diet, leads to an increase in both muscle mass synthesis and muscle mass breakdown. This continuous turnover of muscle proteins results in the efficient repair and renewal of damaged (either mechanically, through oxidation, or otherwise) proteins [37]. On the other hand, exercise training can improve muscle metabolism and ameliorate abnormalities of muscle function without altering the functional performance of the heart [38]. It has been reported that exercise training can increase mitochondrial volume by up to 40% [37]. During physical exercise, the factors that regulate mitochondrial biogenesis are increased, directly enhancing mitochondrial protein synthesis. In aging skeletal muscle, mitochondria are smaller, with slower metabolism and reduced biosynthesis, resulting in a rapid decline in muscle mass and muscle performance parameters [36]. Moderate exercise training can protect mitochondria from volume and biogenesis-induced reductions in volume and biogenesis from aging and therefore attenuate the age-associated decline in skeletal muscle mass [36].

Progressive muscle-strengthening activities maintain or increase muscle mass and strength [39]. This training method is a well-established treatment for muscle atrophy and has been shown to: (a) shorten hospital stays; (b) enhance muscle cross-sectional area; and (c) improve grip strength in older adults [40]. Progressive resistance training is effective for both strength and muscle mass enhancement in older individuals and has been widely used in this population [41,42,43].

According to current evidence, aerobic training has little effect on skeletal muscle size compared to resistance exercise. However, aerobic activity can help slow down muscle loss with aging [39]. Moderate-intensity aerobic exercise, specifically walking, seems to improve the quality of skeletal muscle in healthy elderly adults who already have a sedentary lifestyle by improving the strength of the quadriceps muscles, without, however, increasing lean muscle mass [44].

For the elderly, multiform therapeutic exercise, which is a combination of resistance exercises, aerobic exercises, and balance and/or walking exercises, has also been proposed to improve muscle mass, strength, and functionality [45]. In addition, engaging in strength exercises at least twice a week has been associated with a reduced risk of sarcopenia, greater muscle mass, and better physical performance in elderly adults who already accumulate 150 min of moderate-intensity aerobic exercise per week [46,47]. Combined resistance training and aerobic programs have shown significant beneficial effects on anthropometric characteristics, including muscle mass and functional parameters [48,49,50].

In contrast, physical inactivity and sedentary behavior contribute to low levels of energy expenditure and result in adverse effects, including loss of aerobic capacity and musculoskeletal and cognitive decline [51].

The aim of this review is to summarize the recent research data on the independent and combined effects of diet and exercise and their effect/impact on skeletal muscle mass in the elderly population/in older adults.

## 2. Methods

The PubMed database was searched using the following search strategy: (diet [Mesh] OR nutrition [Mesh] OR vitamin [Mesh] OR exercise [Mesh] OR “physical activity” [tiab]) AND (muscles [Mesh] OR “muscle mass” [tiab]) AND (elder* [tiab] OR old* [tiab] OR aged [Mesh] OR “older adults” [tiab]). The search restricted the articles to the last 5 years (2017–now). 

Inclusion criteria were: (1) older adults > 65 years old; (2) randomized controlled trials; (3) per os supplementation of nutrients; and (4) English language. Furthermore, muscle mass should be measured by BIA, DXA, MRI, or CT. Participants with diseases such as cancer, kidney disease, mobility issues, osteoarthritis, HIV, transplants, liver diseases, Cushing syndrome, arthritis, and neurological diseases (Parkinson) were excluded. Critically ill patients or patients with comorbidities are also excluded. Trials with home-based exercises are not eligible. Finally, we excluded studies in animal models, in vitro, and in vivo. 

## 3. Results

A total of 3510 studies were identified through a PubMed search. Among these studies, 3407 were excluded by the title and/or abstract. In total, 103 studies were screened in full text, of which 41 were included in the review.

Twelve RCTs [41,42,43,44,48,52,53,54,55,56,57,58] examined the effect of exercise on muscle mass, as described in Table 1. Of these studies, one trial performed aerobic exercise as an intervention [44], one performed circuit exercise training [53], and all the rest performed resistance exercise training. The types of resistance exercises differed between studies; one of them used only bodyweight exercises [58], others used elastic bands [52,55] or weights [41,43,56], and others used a combination of the above [42,48,57,58]; only one trial used aerobic training alone [44]. Participants in five of the included studies [43,48,53,54,56] had sarcopenia, whereas only one included post-menopausal women [42], and all the rest trials had healthy older adults. As a measurement method for muscle mass, six trials used BIA [42,48,52,53,55,56,57], one used MRI [41], and two used DXA [44,54].

A total of eleven trials [59,60,61,62,63,64,65,66,67,68,69,70] investigated the role of diet on muscle mass (Table 2). These studies used protein supplementation alone [59,60,61] or combined with other micro- or macronutrients [62,63,64,65,66,67,68,69,70]. Participants in most of the included studies were healthy older adults, except in two where sarcopenic [64,69] or pre-sarcopenic [70] patients participated. DXA [59,60,61,62,63,65,66,69,70] and BIA [64,67] were the methods most commonly used for body composition assessment.

Sixteen trials [49,50,71,72,73,74,75,76,77,78,79,80,81,82,83,84] used a combination approach of diet and exercise training to investigate their role on muscle mass in the third age (Table 3). Only four studies include older adults with sarcopenia [50,73,75,83], whereas the other studies include healthy older adults. The intervention was mainly focused on resistance exercise training combined with protein supplements [49,71,73,74,78,81]. However, four studies supplemented n-3 fatty acids [50,79,80,82] and four used a combination of macro- and micro-supplements [75,77,83,84]. Out of the sixteen included studies, only three performed a combination of resistance exercise training and aerobic training [49,50,75], and only one trial performed aerobic exercise training alone [72]. 

## 4. Discussion

The loss of muscle mass and strength results in a decrease in functionality, which inevitably leads to the inability to perform daily activities, disability, loss of independence, and poor quality of life, leading to the creation of frailty syndrome while at the same time placing a significant burden on the health system. Recent descriptive, synchronic, and prospective studies confirm the importance and magnitude of the problem since the number of falls, fractures, and inability to self-care occur with high frequency in elderly people over 70 years old with sarcopenia [85].

Both older people and elders are at risk of the effects of muscle mass loss with age. Nutritional support for preserving or even enhancing muscle mass is an area of particular interest. Protein intake provides amino acids, which are required for muscle synthesis, as already mentioned. There is also a general concern that older people experience blunted muscle production, which raises the possibility that the overall recommendation for protein intake should be higher for older people [86]. A higher amount of protein (25–30 g) is required to overcome the threshold of maximal activation of muscle protein synthesis [87]. According to ESPEN, protein intake in older people should be at least 1 g/kg/bw/day [27]. The amount should be determined individually and should depend on the nutritional status, the level of physical activity, and the status of any possible existing diseases [27]. The distribution of protein between meals also showed a high frequency of adequate consumption [88]. Data also show that increased protein intake at morning meals led to increased protein intake throughout the day [89] and to the maintenance of skeletal muscle mass [90] in those who consumed increased protein in their evening meal. Timing and frequency of protein consumption are also considered important for muscle synthesis, while it needs to be evenly distributed among the main meals. Based on current evidence, it is observed that to maximize anabolic protein consumption, one should target an intake of 0.4 g/kg/meal, over at least four meals, to reach at least 1.6 g/kg/day [91]. Using the upper daily intake of 2.2 g/kg/day, distributed over the same four meals, a maximum of 0.55 g/kg/meal would be needed [91]. In addition, it has been suggested that muscle protein synthesis is maximized in young adults with an intake of ~20–25 g of high-quality protein. 

Muscle protein synthesis is attenuated in the older and in the elderly when protein and carbohydrates are co-ingested or when the intake of protein is less than about 20 g per meal. The pivotal role of protein supplementation in muscle mass maintenance and development has been widely investigated. Supplementing regular nutrient-mixed meals with leucine may also improve the muscle protein synthesis rate in the elderly [92] Supplementation of soy protein, whey protein, and their combination in older adults had a neutral effect on the maintenance of the appendicular skeletal mass index (ASMI) for a six-month supplementation, compared to baseline status [61]. Furthermore, fortification with whey or collagen peptides in cases of older people with low energy activity and under energy balance, deficit, and recovery conditions revealed that only the whey supplement protein improved leg lean mass and muscle protein synthesis in the recovery period [62]. An uptake in SMM was observed after 3 months of a supplemented diet with a carbohydrate mixture (fat and low protein) in community-dwelling individuals over 70 years of age [67]. The impact of whey protein supplement intake or dietary intake on muscle strength and mass improvement demonstrated no differences between the above treatments. However, supplement intake was found to improve walking speed in older people under 75 years of age. Whey protein provides a mixture of essential amino acids, with leucine being found in high concentrations. In addition to leucine, other amino acids, such as arginine, cysteine, and glutamine, may be involved in the anabolic effects of whey protein supplementation [93]. Six months of dietary supplementation with beta-hydroxyl, beta-methyl butyric acid (HMB), arginine, and glutamine resulted in significant improvements in total lean body mass among healthy older adults [59]. Casein is also digested slowly and progressively during sleep [94]. Ingestion of 30–40 g of casein (milk protein) 30 min before bedtime or via nasogastric tube increased muscle protein synthesis overnight in both young and elderly men [95]. Moreover, dairy products are a good source of high-quality protein and contain various essential nutrients [96]. Dairy proteins, in an amount of 14–40 g/day, can significantly increase the muscle mass of the limbs in middle-aged and elderly adults without, however, having a significant clinical effect on handgrip and leg press muscle strength [97].

Protein supplementation may enhance the effect of exercise on skeletal muscle signaling, particularly by increasing anabolic pathways and decreasing catabolic pathways. The additional effects of protein supplementation on this signaling can be explained by the increased availability of certain amino acids, such as leucine, and dipeptides, including glutamine dipeptides [93]. Hydrolyzed whey protein is a rich source of these ingredients. The way in which leucine induces these anabolic effects within the muscle cell appears to be through stimulation of the mammalian target of rapamycin complex 1 (mTORc1). This is achieved by inducing two main effects on the mTORc1 pathway. The first takes place through the activation of ribosome S6 kinase p70, and the second occurs through the inactivation of eukaryotic initiation factor 4E binding protein 1 (eIF4B), a protein known to repress protein translation. Other key downstream factors of mTORC1 signaling are ribosomal protein S6 (S6) and eukaryotic elongation factor 2 (eEF2), where activation of these proteins will eventually lead to protein synthesis. Intervention with protein supplementation in individuals working in parallel with resistance training resulted in increased expression of S6 and eEF2, indicating a signal favoring protein synthesis, which may be stimulated by increased leucine supplied to skeletal muscle [91]. The whey protein dose effect has been studied in the elderly and showed greater stimulation of muscle protein synthesis with the consumption of 35 g of whey protein, compared to 20 and 10 g of whey protein [98]. Furthermore, lower values of skeletal muscle index, handgrip strength, and performance were associated with lower blood leucine levels [99]. According to recommendations, 3 g of leucine is needed at each main meal to prevent muscle mass loss in older adults [99]. The richest sources of leucine are lean meat, whey products, dairy products, peanuts, lentils, and black beans [99]. Nevertheless, taking supplements makes it easier for this age group to achieve the nutritional goals for their age [64].

On the other hand, branched amino acids and especially leucine supplements’ effects on muscle mass have been studied further. Murphy et al. 2021 [63] observed no effect of leucine supplementation or a combination of leucine with omega-3 fatty acids on muscle protein synthesis, lean mass, and strength in elderly subjects at risk of sarcopenia.

The effect of a combined supplement containing essential amino acids, vitamin D, and a mixture of alanine, resveratrol, CoQ10, and creatine in healthy elderly subjects was evaluated in a 12-week study [65]. The results showed a significant positive effect on muscle mass and strength, covering the age-related loss of years [65], compared to the control group receiving an isothermal maltodextrin supplement. However, this study cannot determine which ingredient or ingredients of the supplement affected the increase in muscle mass, as they were not studied separately.

Another macronutrient compound affecting muscle mass is medium-chain triglycerides (MCTs). MCTs are triglycerides consisting of a glycerol skeleton and three glycerol chains with an aliphatic tail of 6–12 carbon atoms. Abe et al. [68] examined whether intake of medium-chain triglycerides (MCTs) alone is sufficient to increase muscle mass, strength, and function in nursing home elders. MCTs alone or in combination with leucine tended to have a positive effect on the arm muscle area (AMA), increasing compared to the control group. In addition, those effects are reversible within a short period after discontinuation of the intervention. Vitamin D has been linked to the pathogenesis of sarcopenia and other diseases [100]. A study on the combination of vitamin D and protein is adding to the questions about the energy reserves required for the action of vitamin D and protein. In particular, a randomized, controlled, double-blind trial showed a greater increase in muscle mass with supplementation of vitamin D-enriched whey protein and leucine in participants with sarcopenia who had higher initial serum concentrations of 25(OH)D as well as a higher initial dietary protein intake. This suggests that blood vitamin D concentrations greater than 50 nmol/L and a sufficiently high dietary protein intake of more than 1 g per kg bodyweight per day may be required to induce a significant long-term increase in muscle mass from supplemental intake of these macronutrients [69]. The PROVIDE study evaluated the effect of baseline serum 25-hydroxyvitamin D [25(OH)D] and dietary protein concentrations on muscle mass and strength in elderly subjects receiving a dietary intervention. Results showed that basal concentrations of vitamin D and protein were essential for the 3-month dietary intervention to increase muscle mass and muscle strength in the study’s participants [69]. Appropriate baseline levels of 25(OH)D and protein intake may be required to improve muscle mass as a result of vitamin D and protein supplementation intervention in sarcopenic older adults. This implies that current levels of vitamin D and protein intake recommendations could be considered the “minimum” for sarcopenic adults to adequately respond to dietary strategies aimed at mitigating muscle loss. Similarly, concomitant administration of a mixture of leucine, cholecalciferol, and medium-chain triglycerides had a positive effect on increasing appendicular muscle mass (AMM), SMI, and SMM in subjects with high levels of vitamin D at baseline [69]. Additionally, in a six-week intervention study combining vitamin D supplementation with whey protein and leucine at breakfast, muscle mass and protein synthesis in healthy elderly subjects were assessed. The results revealed that this dietary intervention stimulated protein synthesis and increased muscle mass in the elderly in the intervention group, statistically significantly compared to the control group consuming a placebo [66]. Many researchers have studied the effect of combining exercise and diet on maintaining or increasing muscle mass in the elderly [72,73,76,77]. The combination of a nutritional supplement and resistance training had no further improvement in muscle mass. The content of each drink had a total energy of 150 kcal and contained 20.7 g protein (whey protein, 3 g leucine, >10 g essential amino acids), 9.3 g carbohydrates, 3 g fat, vitamins (e.g., 800 IU vitamin D, 2.9 mg vitamin B6, 3 μg vitamin B12), and minerals [101]. Similar results were found after 6 weeks of resistance training with concomitant administration of β-hydroxyl-β-methyl-butyrate acid (HMB), a metabolite of leucine. HMB supplementation showed a marginal increase in lean body mass in the thighs compared to placebo, indicating an early increase in protein synthesis compared to the first two weeks of resistance training [71]. A synergistic effect is observed between protein intake, vitamin D, and exercise [83]. A 12-week randomized controlled trial during which elderly subjects with sarcopenia or dynapenia (low body strength) were exercising with bodyweight resistance training while ingesting a supplement with 10 g of whey protein and 20 μg of vitamin D increased muscle mass at the four extremities in those who had sarcopenia but not in those with normal mass [83]. In contrast, the phase angle, an indicator showing the quality of muscle mass related to sarcopenia [102], was increased in the elderly with low physical function and normal muscle mass but not in those with sarcopenia [83]. A significant beneficial effect of supplemental whey protein and vitamin D intake compared to placebo was found in elderly subjects with sarcopenia who participated in controlled resistance training with a total increase of 1.7 kg in body fat-free mass [99]. Τhe beneficial effects observed only in the supplemented group suggest that physical activity is important but not sufficient to achieve a significant effect. However, it should be noted that physical activity was mild and non-intensive, which may explain the lack of increase in free body fat mass in the placebo group [99].

Another component affecting muscle mass and function is exercise. Resistance training programs appear to have maximal effects on the muscle mass of older adults and elders. Training with elastic bands or/and body weight as a workload at a frequency of three times weekly increased lean body mass (LBM) and lean mass of arms and legs also in pre-sarcopenic subjects 70 years of age [103], and an increase in thigh muscle thickness in men and women as well. In addition, an increase in total quadriceps and triceps arm muscle was found with the application of a resistance training program (speed: 3 s in the concentric phase, 3 s in the eccentric phase) in sedentary elderly men [41]. 

Optimal results in maintaining and regaining muscle mass seem to be achieved through a combination of protein mixtures and exercise types. Particularly, resistance training with a progressive increase in workload and whey protein and leucine supplementation (21 g and 3 g, respectively) led to an augmentation of whole-body lean mass. No change in type I muscle fiber was observed in mass [81]. On the contrary, an elevation of SMI took place after aerobic training in older adults over 65 years old [50] with or without 4 g of n-3 fatty acid intake. Similarly, no statistically significant differences were found in the combined 3 g of n-3 fatty acid intake and lower limb resistance exercise in women over 65 years of age [82]. A combination of 10 g whey protein intake, 800 IU of vitamin D, and resistance training led to an elevation of ALM compared to control subjects [83]. 

Moreover, no effects on quadriceps cross-sectional area were found when carbohydrate intake was compared with protein and collagen intake under high-intensity resistance exercise conditions in community-dwelling older adults. Nevertheless, a significant elevation of quadriceps cross-sectional area was observed in the high-intensity resistance training group compared to the whey protein group [84].

Taking together all the above, regaining muscle mass is a multidimensional field. Adequate protein intake, along with vitamin D and increased physical activity through exercise, are important interventions to mitigate sarcopenia, which may contribute to prolonged independence and vitality in old age.

Protein supplementation and resistance training seem to be the main factors, but not the only ones, for muscle mass growth. The optimal combination has not been found so far. Global dietary recommendations for older adults and elders have not been established to date.

## 5. Conclusions

Changes in body composition associated with aging include a gradual increase in total adipose tissue mass, redistribution of adipose tissue with a preference for muscle fibers and internal organs, and a reduction in peripheral fat. Total skeletal muscle mass reduction is up to 80% from the age of 20 to the age of 80. There is a need to determine the optimal type, amount, timing, and frequency of both the required exercise and dietary intake and supplementation of proteins, amino acids, and vitamins in middle-aged and elderly people and then examine the clinical effectiveness in improving the primary outcomes of sarcopenia. 

By better understanding the impact of physical activity and the importance of nutrient sources, particularly protein and amino acids (>1 g/kg bw/day), on human muscle turnover, it will be possible to develop better strategies and new multidisciplinary approaches, combining exercise and nutrition, to address the physiological age-related loss of muscle mass caused by numerous age-related comorbidities.

The intake of nutrient supplements should also be evaluated based on the individual nutrient needs of each person. As a different mixture of molecules is involved in different stages of muscle metabolism, it will be necessary each time to create an individually specific mixture of ingredients according to human needs. Possible future research should be directed in this direction.

Type, frequency, intensity, and timing of exercise combined with an adequate high-quality protein and micronutrient intake seem to be reliable ways of facing muscle wasting by aging. However, the results are quite controversial and contradictory in many cases, which is probably due to several limitations and variations from study to study, such as the heterogeneity of the samples studied, the way muscle mass is measured, the time and mode of intervention, and comorbidities in the population concerned. Additionally, there is a dose-response relationship between exercise and muscle mass improvement. Time has been considered one of the main factors for muscle adaptation. Thus, a 12-week-duration exercise program [41,42] has the optimal effects on muscle mass, and detraining after three months of participation in an exercise program is unavoidable and takes place in a short period of three weeks. Studies focusing on exercise velocity demonstrate maximal effects on muscle mass regain in elders, while workload plays an important role in muscle mass increase in older people [41,42,43].

There is a need for larger scale and high-quality randomized controlled trials with longer follow-up and standardized primary results (muscle mass, muscle strength, and physical performance), investigating the role of exercise, protein, amino acids, and vitamins in the prevention and treatment of sarcopenia. More randomized control trials are needed to confirm physical activity and diet recommendations to fight muscle loss.

## Figures and Tables

**Table 1 foods-12-01218-t001:** Patients characteristics.

Study, Year	Country	Study Design	Sample Size (TG/CG)	Age	Participants	Sex, Female	Exercise Intervention				Control Group	Muscle Mass	Results
							Type	Characteristics	Frequency	Duration			
Létocar et al. 2021 [41]	France	RCT	MG, *n* = 13; HLG, *n* = 14; Y, *n* =11	MG: 70 ± 4.6; HLG: 69.8 ± 4.4; Y: 24.8 ± 3.6	Sedentary elderly men and active young men	0	RT	10 min warm up in cycle ergometer and 3–5 sets × 15^−4^ reps for muscle groups of the triceps surae and quadriceps with seated calf extension, leg extension and seated leg press (3 s in concentric, 3 s in eccentric phase) MG:55% 1RM HLG:80% 1 RM n3: 14 at 80% 1 RM Y: 55% 1RM	3 d/wk	12 wks	N/A	MRI	Y: ↑ in muscle volume on VL (+5.1%; *p* < 0.05), on VI (+4.8%; *p* < 0.05) and on the total quadriceps volume (+4.3%; *p* < 0.05) after training (quadriceps). ↑ in mean ACSA of the MG and LG muscles on the portions 25% to 50% and 50% to 75% (+3.7% and +8.0%, respectively; *p* < 0.05) and in muscle volume on MG (+3.8%; *p* < 0.05), LG (+8.4%; *p* < 0.05) and on total TS volume (+2.8%; *p* < 0.05) after training (triceps) MG: ↑ in the mean ACSA values of the VL muscle (+6.7%; *p* < 0.05), of the VI muscle on the portions 25% to 50% (+3.6%; *p* < 0.05) and 50% to 75% (5.1%; *p* < 0.01), and of the VM muscle on the portion 0% to 50% (+4.4%; *p* < 0.05) and ↑ in muscle volume on VL (+8.3%; *p* < 0.05), VI (+6.1%; *p* < 0.01), VM (+5.4%; *p* < 0.05), and total quadriceps (+6.7%; *p* < 0.01) after training (quadriceps) ↑ in mean ACSA of the MG and LG on portions 50% to 75% (+10.9% and +14.1% respectively; *p* < 0.01) and also 25% to 50% for LG (+9.0%; *p* < 0.05) and on 75% to 100% for the MG (+6.4%; *p* < 0.05) after training, and in muscle volume on the MG (+10.5%; *p* < 0.05), LG (+14.6%; *p* < 0.05) and total triceps surae volume (+7.5%; *p* < 0.05) (triceps) HLG: ↑ in mean ACSA values of VL muscle on the portions 50% to 75% (+5.2%; *p* < 0.01) and 75% to 100% (+4.0%; *p* < 0.05), in VI and VM muscles on portions 25% to 50% (+5.4% and + 4.1% respectively *p* < 0.05), in LG muscle on portions 25% to 50% (+9.5%; *p* < 0.05) and ↑ in muscle volume on VL (+4.3%; *p* < 0.05), VI (+4.7%; *p* < 0.05), VM (+3.6%; *p* < 0.05), and on the total volume of the quadriceps (+4.2%; *p* < 0.05) after training (quadriceps). ↑ in muscle volume on MG (+8.2%, *p* < 0.05), LG (+9.0%; *p* < 0.05), and in the total volume of TS (4.3%; *p* < 0.05) after the training (triceps) There was no effect on the average ACSA values of the RF muscle and the soleus muscle (*p* > 0.05) regardless of training group.
Osco et al. 2021 [42]	Germany	Longitudinal, two-group, two-time RCT	RT, *n* = 18; EB, *n* = 15	68.7 ± 6.9	post-menopausal older women	All	RT	RT: dynamic constant external load machines; exercises for upper and lower extremities 75 min/session: 2 × 15 to 3 × 12^−15^ to 3 × 8^−12^ 90 sec recovery between sets 20 RM-15 RM EB: major muscle groups	3 d/wk	12 wks	N/A	BIA, DE- XA, BI-VA	↑ in muscle mass after the exercise only in the RT group (31.6 ± 8.1 vs. 34.1 ± 8.6; *p* < 0.05) significant (*p* < 0.05) group by time interaction and time effect for muscle mass.
Bårdstu et al. 2020 [52]	Norway	open label, two-armed, parallel group, cluster randomized trial	64/43	86 (meadian)	Older adults receiving home care	TG: 66%; CG: 51%	RT	Progression RT with elastic bands Rowing, chest press, squats, biceps curl, knee extension, shoulder press (at 3rd phase), up-and-go (at 4th phase) 1st phase: 5 wk, 2 × 10^−12^ reps, 2nd phase: 10 wk, 3 × 10^−12^ reps, 3rd phase: 10 wk, 3 × 8^−10^ reps, 4th phase: 10 wk 4 × 4^−10^ reps 30^−45^ min/session Exercise to fatigue velocity: slow controlled in the eccentric phase	2 d/wk	8 mo	PAR	BIA	No differences in muscle mass between groups were found after 4 or 8 mo of intervention.
Jung, Kim and Park, 2019 [53]	Korea	IRCT	13/13	74.9 ± 4.5	Women with sarcopenia	All	CER	25–75 min; 1–2 wks: 25 min, 3–8 wks: 40 min, 9–12 wk 55 min Main exercise (upper and lower extremities): 10 min followed by 5 min of rest before the next set; program ended with a cool down period of 10 min Intensity: ranged from 60–80% of the HRR	3 d/wk	12 wk	maintain usual physical activity lifestyle	BIA	There were no differences in ASM between the two groups and in the group after the intervention.
Brightwell et al. 2019 [44]	Texas	RCT	12/11	TG: 73,7 ± 4.05; CG: 71.4 ± 4.18	Low active healthy elders	TG, *n* = 8; CG, *n* = 8	AT	Walking in treadmill 45 min at 70% HRR of the HRR	3 d/wk	24 wk	No exercise	DXA	There was no change in total lean mass in TG or in leg lean mass.
Seo, Yang et al. 2021 [54]	South Korea	RCT	12/10	TG: 70.3 ± 5.38; CG: 72.9 ± 4.75	Women with sarcopenia	All	RT	5 min warm-up, 50 min RT, 5 min cool-down; 1 min rest time between sets Main exercises: Upper and Lower body with bodyweight and elastic band progressive increasing of working load during the wks: 6–15 reps, 3–5 sets, 4–8 Omni Scale/Yellow	3 d/wk	16 wk	No exercise	DXA	There was no interaction effect on muscle mass after the intervention. ↑ in follistatin (a muscle growth factor) in the RT (*p* < 0.05).
Chen et al. 2018 [43]	Taiwan	RCT	17/16	TG, *n* = 66.7 ± 5.3 CG, *n* = 68.3 ± 2.8	Sarcopenic Elderly Women	All	RT	Kettlebell weight; exercises for upper and lower body Progressive resistance: full-body major muscle groups: 3× 8^−1^2, 2–3 rests, 60 min/session training with 60–70% of 1 RM	2 d/wk	8 wk	No exercise	Not referred	↑ in ASM at wk 8 and wk 12 than wk 0 (*p* < 0.05) in TG. ↓ in SMM at wk 8 and wk 12 than wk 0 (*p* < 0.005) in CG and the ASM at W8 than wk 0 (*p* > 0.05). There was no difference in SMM during the time at TG.
Urzi et al. 2019 [55]	Slovenia	RCT	11/9	84 ± 8	Nursing home residents	All	RT	10 min warm-up, 35–40 min RT 8 exercises for upper and lower extremities with elastic bands moderate-intensity ERT: Borg Rate of Perceived Exertion scale level “somewhat hard” (12–14), which ranges between “light” (11) and “hard” (15) self-perceived exertion levels.	3 d/wk	12 wk	No exercise	BIA	No significant difference in muscle mass was found within the groups after the intervention, nor between the TG and CG.
Iranzo et al. 2018 [56]	Spain	Parallel group RCT	PMGT, *n* = 11; RMGT, *n* = 9; GC, *n* = 17	PMGT: 87.1 ± 3.8; RMGT: 82.6 ± 9.1; GC: 81.2 ± 5.4	Institutionalized older adults with sarcopenia	PMGT, *n* = 56%; RMGT, *n* = 82%; GC, *n* = 71%	RT	PMGT: ten isotonic resistance exercises ×12 reps, 20–30 min; both concentric and eccentric phases and with two min recovery time between them exercises with dumbbells and ankle/wrist weights for upper and lower body workload adjusted to 40–60% of maximal isometric muscles strength RMGT: using trainer device; workload adjusted to 40–60% of MP	3 d/wk	12 wk	No intervention	BIA	There were no changes in the ASM indices (ASM/height^2^, ASM/weight, and ASM/BMI) between pre- and post-intervention in the three groups.
Piastra et al. 2019 [57]	Italy	RCT	RTG, *n* = 33; PTG, *n* = 33	RT: 69.9 ± 2.7 y.o PTG: *n* = 37, 70.0 ± 2.8 y.o.	Community dwelling older women	All	RT or Postural	60 min/session RT: 15 min warm-up, 30 min; low/moderate intensity for different muscles (abdominal and both lower and upper limbs); low weight (0.5, 1, or 1.5 Kg) and 15 min cool-down Postural training: 15 min cardiovascular activation, upper and lower body mobilization, 10–15 min neck and shoulders mobilization and 5–30 min spine mobilization, and exercises for stretching, and final relaxation	2 d/wk	36 wk	N/A	BIA	↑ in SM and SMI in the RTG (T0 = 17.31 ± 1.16 kg, T1 = 19.02 ± 6.58 kg, *p* < 0.001 and T0 = 6.48 ± 2.75 kg/m^2^, T1 = 7.36 ± 2.31 kg/m^2^, *p* < 0.001). No significant differences appeared in the PTG.
Lee et al. 2019 [58]	Korea	RT	Young-old, *n* = 67 (TG, *n* = 31; CG, *n* = 36) Old-old, *n* = 69 (TG, *n* = 32; CG, *n* = 37)	74.54 ± 6.37	community-dwelling older adults	122	RT	Upper and lower body exercises were performed in separate day; 3–5 × 15^−20^ Upper body: (a) bodyweight exercises without instrument, (b) elastic bands for upper body Lower body exercises: bodyweight Defined by e Borg Rating of Perceived Exertion scale: 15	2 d/wk	8 wk	Maintain current lifestyle	Not reffered	↑ in SMM after exercise intervention in the young-old group (*p* < 0.025), whereas changes in SMM were not observed in the old-old group.
Flor-Rufino et al. 2023 [48]	Spain	Single blind RCT	20/18	79.8 ± 7.4	community-dwelling older women with sarcopenia	All	HIRT	65 min/session: 10 min warm up, 40–45 min, circuit HIRT, 10 min cool-down 6 exercises to strengthen different muscle groups (2 upper body, 2 on the trunk and 2 on the lower body). lower body exercises were only leg press and knee extension. 3 × 10^−15^; 70% of 1RM	2 d/wk	6 mo	No intervention	BIA	↑ in muscle mass (+1.1 kg; *p* < 0.05) and SMI (+0.4 kg/m^2^ *p* < 0.001) within-group analyses significant group × time interaction effect for muscle mass (*p* = 0.027; Ƞ^2^ = 0.129) and muscle mass index (*p* = 0.023; Ƞ^2^ = 0.135) for HIRT.

TG = treatment group; CG = control group; m = month(s); wk(s) = week(s); d = day(s); MG = moderate group; HLG = high-load group; Y = young group; RT = resistance training; RM = repetition maximum; MRI = magnetic resonance imaging; VL = vastus lateralis; VI = vastus intermedius; ACSA = appendicular cross-sectional area; MG = medial gastrocnemius; LG = lateral gastrocnemius; TS = triceps surae; VM = vastus medialis; RF = rectus femoris; EB = elastic band group; BIA = bioelectrical impedance analysis; DXA = dual-energy X-ray absorptiometry; BI-VA = bioelectrical impedance vector analysis; PAR = physical activity recommendations; reps = repetitions; HRR = heart rate reserve; CER = circuit exercise training; ASM = appendicular skeletal mass; AT = aerobic training; SMM = skeletal muscle mass; PMGT = peripheral muscle training group; RMGT = respiratory muscle training group; PTG = postural training group; SM = skeletal muscle; SMI = skeletal muscle index; HIRT = high-intensity resistance training.

**Table 2 foods-12-01218-t002:** Studies that examined the effect of diet on increasing muscle mass in the elderly.

Study, Year	Country	Study Design	Sample Size (TG/CG)	Age	Participants	Sex, Female	Intervention			Control Group	Muscle Mass	Results
							Supplement	Dose	Duration			
**Protein Supplementation**												
Ellis et al. 2019 [59]	USA	Double-Blind	16/15	>65	Community dwellings	17	Amino Acid - HMB, - L-arginine - L-gluta- mine	2 times/day with food, 3 g HMB, 14 g L-arginine and 14 g L-glutamine	6 m	2 times/day with food, powdered placebo	BODPOD DXA	↑ in LM in the intervention group (*p* = 0.003) but not in the placebo group (*p* = 0.688). ↑ in arm LM only in the intervention group (*p* = 0.011). ↑ in leg LM within both the intervention (*p* = 0.024) and control group (*p* = 0.025). ↑ in FFM within the intervention group (*p* = 0.012).
Ten Haaf et al. 2019 [60]	The Netherlands	Double-blind, controlled intervention study	58/56	69	Physically active older adults	18	PRO	2 packs of supplement: 36.8 g milk PRO with 31 g PRO, 1.1 g fat, 14.5 g lactose (CHO) 1 pack during breakfast and the 2nd pack 30 min after exercise (e.g., walking). On non-exercising days, during lunch.	12 wk	250 mL iso-caloric placebo drink/twice per day 500 mL of supplement: 1.1 g PRO, 5.2 g FAT, and 36 g of CHO	DXA	↑ in whole-body LM in the protein group had a greater effect than the placebo group (Δ0.93 ± 1.22% vs. Δ0.44 ± 1.40%, *p* Interaction = 0.046) ↑ Truncal lean body mass in the protein group compared with the placebo group (*p* Interaction = 0.007).
Li et al. 2021 [61]	South China	Four arm	WP = 31; SP = 31; WS = 31; *p* = 30	70 ± 4	Community dwellings	50%	Whey Protein Soy Protein Whey-soy Protein	16 g/d, 2 times/d	6 m	No supplement	DXA	There were no changes in ASMI or lean mass in the legs, arms, trunk, and appendicular areas in the supplemented groups; these values decreased from baseline in the control group (*p* < 0.01).
**Combined supplementation**												
Oikawa et al. 2018 [62]	Canada	Double-blind, parallel-group	16/15	69 ± 4/68 ± 2	Community dwellings	16 M 15 F	Amino Acids, Collagen Peptide All consume a total of 1.6 g protein/d	30 g of supplements, 2 times/d	5 wk	N/A	DXA	↓ in LBM during energy restriction and supplement consumption in comparison to energy balance with protein intake equal to the RDA (0.8 g protein kg^−1^ d^−1^) period (*p* < 0.001). ↑ in LBM during the recovery period only in the whey protein group.
Murphy et al. 2021 [63]	Ireland	Three arm parallel double-blind	38/38/31	70 ± 5/73 ± 6/73 ± 7	Community dwellings	55	Protein (leucine) with/without LCn-3 PUFA	21.2 g protein/d, which included 6.2 g leucine/d, with or without 4 g LC n–3 PUFAs/d	24 wk	Isoenergetic supplement (maltodextrin, sunflower and corn oil)	DXA	There were no differences between CON and either LEU-PRO or LEU-PRO + n–3.
Lin et al. 2020 [64]	Taiwan	Open label, parallel group	28/28		Sarcopenic patients	16	Whey protein vitamin D	12.8 g PRO (8.5 g whey PRO concentrate), 1.2 g leucine, 7.3 g CHO, 0.8 g fat and 120 IU vitamin D per serving.	12 wk	Follow diet with 1.5 g PRO/Kg BW/d	BIA	↑ in AMM at 4 and 12 weeks within the both groups. There were no significant differences in AMMI (*p* = 0.87 at week 4; *p* = 0.3 at week 12) between the 2 groups.
Negro et al. 2019 [65]	Italy	Double-blind	19/19	69.9 ± 4.6	Healthy elders	30	Amino Acids, creatine, vitamin D and Master Restore Complex (ALA, CoQ10, resveratrol)	200 mL × 2/d 5000 mg EAA, 1500 mg creatine, 1000IU VitD, 300 mg ALA, 50 mg CoQ10, 50 mg resveratrol per 200 mL	12 wk	Isocaloric maltodextrine	DXA	statistically significant increase in ALM: +0.34 kg and ALM/H2: +0.12 kg/m^2^ ↑ legs FFM (MD: (mean dif) +443.70 g; *p* < 0.05), ↑ ALM (MD: +0.53 kg; *p* < 0.05) and ↑ ALM/H2 (MD: +0.19 kg/m^2^; *p* < 0.05) between the two groups.
Chanet el al. 2017 [66]	France	Placebo-controlled, double-blind	12/12	71 ± 4	Healthy older men	0	Whey PRO + VitD	21 g leucine enriched whey protein, 9 g CHO, 3 g FAT, 800 IU cholecalciferol before breakfast	6 wk	Flavored watery placebo	DXA	↑ ALM in the TG compared with the CG (ED: 0.37 kg; 95% CI: 0.03, 0.72 kg; ANCOVA, *p* = 0.035) ↑ in leg lean mass (ED: 0.30 kg; 95% CI: 0.03, 0.57 kg; ANCOVA, *p* = 0.034). No important between group differences in lean body mass and in arm lean mass.
Chew et al. 2020 [67]	Singapore	Double-blind parallel design multi-center	401/404	74.1 ± 0.26	Community dwellings	485	ONS and dietary counseling	2 servings of 10.5 g protein, 8.5 g fat, 34.2 g carbohydrate, 310 IU vitamin D3, and 0.74 g calcium HMB per serving and dietary counseling	180 d	Placebo supplement contained 60 kcal, 1.07 g protein, 1.21 g fat and 11.9 g carbohydrate per serving and dietary counseling	BIA	↑ in appendicular skeletal muscle mass (ASM) at day 90, in the intervention group with normal ASMI than a placebo (16.40 ± 0.54 kg vs. 15.52 ± 0.44 kg; *p* = 0.036).
Abe et al. 2019 [68]	Japan	Single-blinded	21/21/22	85.5 ± 6.8	Nursing home elders	51	l-leucine, cholocalciferol, MCT, LCT	group a: l-leucine (1.2 g), cholecalciferol (20 μg) and 6 g/d of MCTs; group b: 6 g/d of MCTs	3 m	6 g/d of long chain triglycerides (negative control)	Triceps skinfold thickness	There was no significant difference in the skeletal muscle in the group or the group-by-time interaction; AMA tended to increase after the intervention in a and b groups, whereas it decreased in the negative control group.
Verlaan et al. 2018 [69]	Europe	Multi-center, double- Blind	380	>65 y	Community dwellings with sarcopenia		25(OH)D Protein (whey, leucine)	2 times/day, 20 g whey protein, 3 g total leucine, 9 g carbohydrates, 3 g fat, 800 IU vitamin D, and a mix of vitamins, minerals, and fiber	13 wk	2 times/day, contained carbohydrates, fat and some trace elements, (no vitamin D)	DXA	↑ in appendicular muscle mass, SMI, and relative appendicular muscle in the high baseline 25(OH)D concentration group compared with participants with 25(OH)D <50 nmol/L ↑ in appendicular muscle mass, SMI, and relative appendicular muscle mass in the group with a higher baseline protein intake (>1 g/Kg/d) compared with the group with a low protein intake (<1 g/Kg/d).
Hajj el et al. 2017 [70]	Lebanon	Parallel group	60/55	73.3 ± 2.05	Pre-sarcopenic elders	66	Vitamin D	10,000 IU cholecarciferol	6 m	Placebo (microcrystalline cellulose = 66.3%, starch = 33.2%, magnesium stearate = 0.5%, per serving)	DXA	↑ in ASMM (kg) from baseline (21.58 ± 6.53 kg) to 6 m (22.23 ± 5.85 kg) in vitamin D group (*p* = 0.001). There was no significant change from baseline (16.83 ± 3.11 kg) to 6 m (16.92 ± 3.25 kg) (*p* = 0.203) in the placebo group. The mean percent change between the vitamin D group (3.01 ± 2.38) and placebo group (0.46 ± 3.30) was significantly different (*p* ≤ 0.001).

TG = treatment group; CG = control group; m = month(s); wk(s) = week(s); d = day(s); HMB = β-hydroxy-β-methylbutyric acid; DXA = dual-energy X-ray absorptiometry; LM = lean mass; FFM = fat-free mass; 25(OH)D = 25-hydroxyvitamin D; SMI = skeletal muscle index; ASMI = appendicular muscle mass index; LCn-3 PUFA = long-chain n-3 polyunsaturated fatty acids; PRO = protein; LEU-PRO = leucine-protein; CHO = carbohydrates; AMM = appendicular muscle mass; AMMI = appendicular muscle mass index; ALA = alanine; CoQ10 = coenzyme Q10; EAA = essential amino acids; VitD = vitamin D; ALM/H^2^ = appendicular lean mass/height^2^; FFM = fat free mass; MD = mean difference; ALM = appendicular lean mass; ONS = oral nutritional supplements; Ca-HMB = calcium- β-hydroxy-β-methylbutyric acid; BIA = bioelectrical impedance analysis; Δ-alm = difference-appendicular lean mass; MCT = medium-chain triglycerides; LCT = long-chain triglycerides; ASMM = appendicular skeletal muscle mass.

**Table 3 foods-12-01218-t003:** Studies that examined the effect of diet and exercise on increasing muscle mass in the elderly.

Study, Year	Country	Study Design	Sample Size (TG/CG)	Age	Participants	Sex, Female	Exercise Intervention				Nutritional Intervention			Control G	Muscle Mass	Outcome
							Type	Characteristics	Frequency	Duration	Supplement	Dose	Duration			
Din et al. 2018 [71]	UK	Double blind control placebo trial	8/8	TG: 67.8 ± 1.1 CG: 68.5 ± 1.6	Healthy adults	0	RT	All study groups performed activities leg extension of the dominant leg 6 sets of 8 reps, 75% 1-RM	3 days per wk	6 wk	HMB-FA	3 g HMB-FA/d Consumed the supplement daily at breakfast	6 wk	Placebo with same flavor and taste	DXA	↑ in lean muscle mass in the trained leg in the HMB-FA group (5734 ± 245 g at 6 wk HMB-FA vs. 5644 ± 323 g PLA, *p* < 0.06; *p* < 0.05 in the HMB-FA group).
Markofski et al. 2018 [72]	Texas	2 × 2 factorial design, double-blind, placebo-controlled	Ex + S, *n* = 14; Ex + PLA, *n* = 11; S, *n* = 13; PLA, *n* = 12	72 ± 1	Health, low active, normal to mildly obese	30	progressive AT	walked on a treadmill for 45 min at 70% HRreserve + 5-min cool down; target of 5% heart rate during the exercise session	3 nonconsecutive days per wk	24 wk	EAA	15 g of EAA/d	24 wk	Placebo	DXA	No change in total and leg lean mass was observed in any group after the intervention.
Mori and Tokuda, 2022 [74]	Japan	RCT	Ex + S, *n* = 23; S, *n* = 23; Ex, *n* = 23	Ex + S: 77.7 ± 3.3; S: 77.8 ± 4.5 Ex: 77.6 ± 5.2	Older adults with sarcopenia	75	RT	Bodyweight: lower body exercises included rising and sitting from a chair, and leg extensions Resistance exercise: elastic band included upper and lower body exercises 50–70% of the 1 repetition maximum, 2–3 sets, 30–40 min per session	2 days per wk	24 wk	PRO	Supplement: 160 kcal, 11 g PRO, 2.2 g FAT, 24 g CHO, and 2300 mg of leucine/serving Ex + S group: 5 min after the completion of the RT program S group: 3 h after the lunch	24 wk	N/A	BIA	↑ in ASMI in 24 wks in the RT + S group (*p* < 0.01). ΔASMI was higher in the RT + S group than in the RT group at 24 wks of de-training (*p* < 0.05). No significant differences in ΔASMI between each group. There was significant group-by-time interaction for ASMI at 24 wks of de-training (*p* = 0.014).
Mori and Tokuda, 2018 [73]	Japan	Open-label, parallel-group	Ex + S, *n* = 25; S, *n* = 25; Ex, *n* = 25	Ex + S: 70.6 ± 4.2; S: 70.6 ± 4.2 Ex: 70.6 ± 4.2	Healthy older women	All	RT	Bodyweight and band exercises Bodyweight: lower body exercises included rising and sitting from a chair, and leg extensions. Elastic band exercises: upper and lower body exercises (seated chest press, seated row, knee extension, squats, knee-ups); 5 diferrent RT levels The resistance load (50–70% of the 1 RM)	N/A	24 wk	PRO	Supplement contained: 92 kcal, 22.3 g PRO, 0.3 g of FAT, 0.1 g of CHO, 1225 mg of valine, 2975 mg of leucine and 1175 mg of isoleucine/25 g of one intake serving Ex + S: 5 min after the RT S: 3 h after lunch	24 wk	N/A	BIA	↑ in lower limb muscle mass and SMI for the Ex group than for the S group (lower limb muscle mass, *p* = 0.018, SMI, *p* = 0.008). ↑ in lower limb muscle mass and SMI for the Ex + S group than for both the Ex (lower limb muscle mass, *p* = 0.038, SMI, *p* = 0.007) and S (lower limb muscle mass and SMI, *p* < 0.001) groups. ↑ in upper limb muscle mass in the Ex + S group than for the S group (*p* = 0.029). Lower limb muscle mass and SMI: significant group by time interaction (*p* < 0.001).
Osuka et al. 2017 [49]	Japan	Open-labeled RCT	Ex + S, *n* = 28; Both type Ex + S, *n* = 28	RT = 70.6 ± 4.0; AT = 69.6 ± 3.5	Healthy adults	RT = 18; ART = 20	RT or both AT and RT	RT: upper and lower body in machines 1 RM in each 4-wk period (baseline, 4 wks, 8 wks, and post-intervention) and increased gradually the intensity (1–4 wks: 30–50% of 1 RM, 5–8 wks: 50–70% of 1 RM, 9–12 wks: ≥ 70% of 1 RM) 1st 4-wk training: 3 sets of 10 reps per set, 5–12 wks: 3 sets of 12 reps per set; rest periods: 1 min AT: in ergometer Intensity: light (40–50% of VO2peak) training volume: gradually increased (1–4 wks: 20 min, 5–8 wks: 25 min, and 9–12 wks: 30 min). 10 min of warming up, 45–60 min of RT or both AT and RT and 10 min of cool down	2 nonconsecutive days	12 wk	PRO fortified milk	10.5 g PRO, 3.9 g FAT, 9.3 g CHO, 87 mg of sodium, and 337 mg of Ca every day after training	12 wk	N/A	DXA	RT group: ↑ in SMI (before: 6.8 ± 1.0 vs. post: 6.9 ± 1.0), whole-body muscle mass (before: 39,795 ± 7882 vs. post: 40,351 ± 8033), upper (before: 3949 ± 1158 vs. post: 4021 ± 1119), and lower (before: 12,651 ± 2784 vs. post: 12,858 ± 2785) extremity muscle mass (all *p* < 0.05). AT group: ↑ in lower extremity muscle mass (before: 13,062 ± 2997 vs. post: 13,199 ± 2974; *p* < 0.05). There were no significant differences in the changes in muscle mass between the two groups.
Zhu et al. 2019 [75]	China	Prospective parallel group, single-blind	ExG: 40; Ex + S: 36; CG: 37	ExG: 74.5 ± 7.1; Ex + S: 74.8 ± 6.9; CG: 72.2 ± 6.6	community-dwelling older adults with sarcopenia	ExG: 72.5%; Ex + S: 80.6%; CG: 78.4%	RT + AT	5–10 min warm-up and cool-down routine, 20–30 min chair-based RT with elastic bands, and 20-min AT RT with bands: muscle groups in both the upper and lower body 6–8 reps, 6 sets submaximal 6–8 RM test for each set of exercise, 40% of the estimated 1RM	2 per wk and 1 home based	12 wks 12–24 wks home exercise sessions	PRO + HMB + VitD + n-3 FA	Two sachets whereas each sachet contained: 54.1 g powder of 231 Kcal, 8.61 g PRO, 1.21 g HMB, 130 IU VitD and 0.29 g n-3 FA	12 wks	Maintain their usual physical activities and dietary habits for 6-mo	DXA	Improvement in lean muscle mass, especially in lower limbs (*p* = 0.015) and in ASM/h^2^ (*p* = 0.025) was only observed in the combined exercise program and nutrition supplement group. Such an increment was not maintained until the 24th week.
Osuka et al. 2021 [76]	Japan	Double-blind, placebo-controlled, 2 × 2 factorial design trial	Ex + S = 36; Ex + PLA = 38; Ed + S = 37; Ed + PLA = 38	65–79	community-dwelling older adults	All	RT	5 min stretching, 50 min RT as the main exercise, and 5 min stretching as a cool-down 1–12 wks: chair-based exercises; 5–7 wks elastic band; 7–12 wks ankle weight, and 9–12 wks machine-based RT 1–3 sets of 8–10 reps with gradual loading. exercise intensity of 12–14 points on the Borg Rate of Perceived Exertion Scale	2 nonconsecutive days	12 wks	Ca-HMB	Active products with 3.5 g CHO, 30 mg PRO, 20 mg FAT, 0.2 mg Na, 207 mg Ca, and 1200 mg HMB/day after any meal	12 wks	Placebo without Ca-HMB	DXA	ITT analyses: no significant exercise × HMB interactions or main effects of exercise and HMB supplementation on muscle mass. Per-protocol analyses: ↑ of upper-extremity lean mass with HMB supplementation by 0.06 kg (*p* = 0.019).
Seino et al. 2018 [77]	Japan	Two-arm, randomized, controlled trial	Ex + S = 40; Ex = 40	73.5	community-dwelling older adults	Ex + S = 85%; Ex = 82.5%	RT	10-min warm-up, 45 min of RT and 5 min of cool-down Upper-extremity training, lower-extremity training and trunk training: Resistance, reps, sets, and speed of movements were increased progressively using weight bearing, resistance bands or Pilates balls. 2 sets/20 reps Target intensity: approximately 5–7	2 per wk	12 wks	PRO + miconutrients	PRO fortified milk: 114 kcal energy, 10.5 g PRO milk, 3.9 g FAT, 9.3 g CHO, and 337 mg calcium per 200-mL pack, at lunchtime, every day micronutrient beverage with 45 kcal energy, 10.5 g CHO, 7200 μg β-carotene, 200 IU Vit D, 30 mg Vit E, 3.0 mg, Vit B1, 4.0 mg Vit B2, 32 mgNE niacin, 6.0 mg Vit B6, 12 μg Vit B12, 200 μg folic acid, 12 mg pantothenic acid, 320 mg Vit C, 7.5 mg Fe, 8.0 mg Zn, 0.40 mg Cu, and 25 μg Se per 125-mL pack at breakfast every day	12 wks	N/A	DXA	↑ in WBLM (0.63 kg (95% CI: 0.31–0.95, *p* < 0.001), ALM (0.37 kg (95% CI: 0.16–0.58, *p* = 0.001), and LLM 0.27 kg (95% CI: 0.10–0.46, *p* = 0.001) in the supplementation group.
Hamarsland et al. 2019 [78]		Double-blinded, randomized, controlled trial	38	Milk group = 74.3 ± 3.6; WheyPRO = 72.9 ± 1.8		6 per group	RT	All study groups performed RT Whole body exercises Loads ranged from 12 to 6 RM, for 1 to 3 sets and progressed from higher to lower repetition ranges Mondays: 1–2 sets of 12 RM for the 1st 3 wks before adding another set to several exercises in 4–9 wks and again in 10–12 wks. Fridays: progressed from 1–2 sets of 8 RM in 1–6 wks to 2–3 sets of 6 RM in 7–12 wks. Mondays and Fridays, workouts were conducted with maximal training load and intensity for the given reps. Wednesdays, workouts were submaximal, using 90% of the load on the previous Monday for the same amount of reps. Inter-set rest periods lasted for 2–3 min	3 days per wk	11 wks	EAA	40 g (2 daily servings of 20 g/serving) No training days: in the morning and in the afternoon Training days: one serving after training and one serving in the afternoon Milk vs. Native Whey (per serv): Ala 1 vs. 0.6, Arg 0.6, Asp 2.2 vs. 1.5, Cys 0.5 vs. 0.2, Phe 0.9, Glut 3.9 vs. 4.1, Gly 0.4, His 0.5, Iso 1.1 vs. 1, Leu 2.5 vs. 1.9, Meth 0.5, Prol 1.3 vs. 1.9, Ser 1 vs. 1.1, Thre 1 vs. 0.8, Tyr 0.7 vs. 0.8, Val 1.2, Try 0.4 vs. 0.2, PRO 21.8 vs. 19.7, FAT 20 vs. 19.1, CHO 7.5 vs. 6.9	11 wks	N/A	DXA Immunohistochemistry	↑ in muscle hypertrophy for lean mass (milk: 6.3 ± 3.6%; native whey: 4.6 ± 3.4%, *p* < 0.001 for both groups) and type II muscle fiber cross-sectional area (milk: 34.2 ± 56.7%, *p* = 0.021; native whey: 33.8 ± 27.4%, *p* < 0.001) in both groups. No change for type I muscle fibers (milk: 4.7 ± 22.0%, *p* = 0.62; native whey: 2.8 ± 19.7%, *p* = 0.89). There were no between-group differences in muscle growth.
Brook et al. 2021 [79]	UK	Double-blind	8/8	S:64 ± 1; PLA:67 ± 1	Healthy older women	All	RT	6–8 reps, 75% 1-RM for the dominant leg (one leg remained untrained throughout the intevention)	3 d/wk	6 wks	n-3 FA	3680 m g/d (1860 mg EPA; 1540 mg DHA)	6 wks	Cornoil	DXA Immunoblotting	↑ in type II muscle fibers in n-3 PUFA after 6-weeks of RET (4329 ± 264 mm^2^ (*p* < 0.05, MD = +1232, 95% CI = [205,2258], d = 1.44; no changes in PLA group. There were no changes in type I muscle fibers.
Cornish et al. 2018 [80]	Canada	Pilot study-based design was a 2-group randomized trial	S: 11; CG:12	S: 71.4 ± 6.2; CG:70.9 ± 5	Old male	0	RT	All participants participate in exercise program with whole body training using weights completed a progressive RT program devided into 4 blocks of 3 wks/block. 1st block: familiarization exercise and anatomical adaptation; 2–3 sets of 10–12 reps at 60–65% of 1RM. 2nd block: developing skeletal muscle hypertrophy by using a higher volume of training; 3–4 sets of 8–12 reps at 65–75% of 1RM. 3rd block: development of skeletal muscle strength; 3–4 sets of 6–10 reps at 70–85% of 1RM. 4th block: in 12-wk; maintenance phase of RT; 3 sets of 8–12 reps at 65–80% of 1RM.	3 days per wk; 48 h rest between training sessions	12 wks	n-3 FA	3.0 g of a combined EPA/DHA (EPA was 1.98 g and DHA was 0.99 g) supplement on a daily basis 3 caps every morning with breakfast.	12 wks	3.0 g of an omega 3–6–9 blend with 45% α-linolenic acid (1350 mg), 26.5% linoleic acid, and γ-linolenic acid (795 mg); 17.5% oleic acid (525 mg); 11% (330 mg) other SC-FA, sat FAT and phospholipids	DXA	↑ in LTM from baseline to 12 wks (55.5–56.1 kg; *p* = 0.032), There was no significant time × group interaction. There was no significant main effect of the group.
Holwerda et al. 2018 [81]	Nehterlands	Double-blinded	S, *n* = 21; E + PLA, *n*= 20;	70 ± 1	normoglycemic older men	0	Whol e- body RT	All participants participate in exercise program 5-min warm-up on a cycle ergometer, and 4 sets on both the leg press and leg extension machines. Upper body exercises were paired and were performed in an alternating manner between training sessions; 2 sets/exercise After training, 5 min cool-down on the cycle ergometer 1st 4wks of training, the workload was increased from 70% 1RM (8 reps/set) to 80% 1RM (10 reps). Resting periods of 2 to 3 min between sets and exercises, respectively	3 d/wk	12 wks	PRO	21 g whey protein enriched with 3 g leucine; each night before sleep, including rest days	12 wks	Energy-matched placebo	DXA Immunohistochemistry.	↑ in whole-body lean mass with RT in both groups (*p* < 0.001); no differences between groups. ↑ in ALM and leg lean mass with RT in both groups (*p* < 0.001); no differences between groups. ↑ in quadriceps muscle CSA in response to 12 wk of RT (*p* < 0.001); no differences between groups. ↑ in type II muscle fiber CSA in response to 12 wk of RT in both groups (*p* < 0.001); no differences between groups. There were no changes in type I muscle fiber CSA in response to 12 wk of RT in both groups.
de Cruz Alves et al. 2022 [50]	Sao Paolo	Double-blind, Placebo-controlled trial	Ex + S, *n* = 16; Ex + PLA, *n* = 16	>65 y	healthy older adults with sarcopenia	all	AT + RT	All participants participate in exercise program Aerobic warm-up session was performed for 10 min. 3 series of 12 reps for each exercise, with 1 min of rest between each series. Both eccentric and concentric phases included 1st 2-wks at 50% of 1RM; 3rd wk, intensity increased to 70% of 1RM and, from the 7th wk to 80% of 1RM	3 d/wk	7 wks	EPA + DHA	4 g/d; two capsules at lunch and dinner	7 wks	Sunflower oil	MRI	↑ in SMI in both groups post-intervention. ↑ in quadriceps CSA after the intervention in both groups (*p* = 0.006); ↑ of 6.11% (from 3.76 cm^2^ to 3.99 cm^2^) in the Ex + S group; ↑ of 2.91% (from 3.44 cm^2^ to 3.54 cm^2^) in the Ex + PLA group; no statistically significant difference between groups (*p* = 0.23).
Da Boit et al. 2017 [82]	UK	Double-blind trial	Ex + S: *n* = 23 (women = 10); Ex + PLA: *n* = 27 (women = 13)	Men = 70.6 ± 4.5; Women = 70.7 ± 3.3	Older adults	23	RT	4 sets of 9 reps for lower-body exercises Intensity for each exercise was set at 70% of the participants’ 1RM	2 d/wk	18 wk	n-3 FA	Three capsules long-chain n–3 PUFAs/d (3 × 1 g capsules giving 2.1 g EPA/d + 0.6 g DHA/d)	18 wk	placebo (safflower oil: 3.0 g/d)	MRI	No group, sex, or interaction effects were seen for muscle ACSA.
Yamada et al. 2019 [83]	Japan	Four-arm randomized controlled trial	Ex + S, *n* = 28; Ex, *n* = 28; S, *n* = 28; CG, *n* = 28	Ex + S = 84.9 ± 5.6; Ex = 84.7 ± 5.1; S = 83.2 ± 5.7; CG = 83.9 ± 5.7	Sarcopenia or Dynapenic older adults	Ex + S = 71.4%; Ex = 64.3%; S = 71.4%; CG = 53.6%	RT	5 min of warm-up activity, 20 min of the RT and 5 min of cool-down activities. 3 sets of 20 reps for each exercise at a slow movement speed using bodyweight or an elastic band	2 d/wk	12 wks	PRO + VitD	Protein and vitamin D supplements were provided every day 100 kcal; 10 g of whey protein + 800 IU of Vit D/day after breakfast	12 wks	None intervention	BIA	↑ in ALM in the Ex + Nutr group than the CG (*p* < 0.05).
Mertz et al. 2021 [84]	Denmark	RCT	CHO group, *n* = 34; COLL, *n* = 44; WHEY, *n* = 44; LITW, *n* = 30; HRTW, *n* = 32	CHO group: 69.6 ± 3.9 COLL: 70.4 ± 4.1; WHEY: 70.3 ± 4.3; LITW: 70.4 ± 4.0; HRTW: 70.3 ± 3.1	community-dwelling adults	CHO group = 18; COLL, *n* = 234; WHEY, *n* = 22; LITW, *n* = 18; HRTW, *n* = 18	RT	HRTW group: 3 exercises for the lower body and 2 upper-body exercises/∼1 h session. 3-mo cycles, increasing the load progressively from 3 sets of 12 reps at a 12 RM to 5 sets of 6 reps at a 6 RM in each cycle. LITW group: performed light-load home-based RT 3–5 times/wk, using rubber bands and their body weight for exercises chosen to mimic the muscle groups and movement used in training those assigned to HRTW	3 d/wk	1 y	CHO or PRO or collage	CHO group: 2 × 20 g maltodextrin + 10 g sucrose/d); Whey group: 2 × 20 g whey protein hydrolysate + 10 g sucrose/d; COLL group: 2 × 20 g bovine collagen protein hydrolysate + 10 g sucrose/d; LITW group: 2 × 20 g whey protein hydrolysate + 10 g sucrose/d); HRTW group: 2 × 20 g whey protein hydrolysate + 10 g sucrose/d	1 y	N/A	DXA MRI	There were no between-group differences in changes in qCSA (time∗group interaction, *p* = 0.17) in the supplementation-only analysis, but the time∗group interaction combining training and supplementation analysis, term was significant (*p* = 0.04) HRTW: ↑ in qCSA compared to WHEY (mean between-group difference, +1.68 cm^2^; 95% CI, +0.41 to +2.95 cm^2^; *p* = 0.03), but not compared to LITW (mean between-group difference, +1.29 cm^2^; 95% CI, −0.08 to +2.67 cm^2^; *p* = 0.16). No changes in qCSA between LITW and WHEY (mean between-group difference, +0.39 cm^2^; 95% CI, −0.88 to +1.66 cm^2^; *p* = 0.82). Within-group changes in qCSA: no changes in HRTW (0- to 12-mo change, +0.73 cm^2^; 95% CI, −0.32 to +1.77 cm^2^) or LITW (0- to 12-mo change, −0.54 cm^2^; 95% CI, −1.70 to +0.62 cm^2^). ↓ in qCSA in WHEY (0- to 12-mo change, −0.93 cm^2^; 95% CI, −1.65 to −0.21 cm^2^).

TG = treatment group; CG = control group; RT = resistance training; RM = repetition maximum; Reps = repetitions; HMB-FA = β-hydroxy-β-methylbutyric acid free acid; m = month(s); wk = week(s); d = day(s); y = year(s); DXA = dual-energy X-ray absorptiometry; Ex + S = exercise + supplement group; Ex + PLA = exercise + placebo group; S = supplement group; PLA = placebo group; AT = aerobic training; EAA = essential amino acids; RCT = randomized controlled trial; Ex = exercise group; PRO = protein; CHO = carbohydrates; BIA = bioelectrical impedance analysis; ΔASMI difference in appendicular skeletal muscle mass index; ASMI = appendicular skeletal muscle mass index; SMI = skeletal muscle index; Ca = calcium; VitD = vitamin D; n-3 FA = n-3 fatty acids; ASM/H^2^ = appendicular skeletal mass/height^2^; Ed + S = education and supplement group; Ed + PLA = education and placebo group; Ca-HMB = calcium-β-hydroxy-β-methylbutyric acid; Na = sodium; ITT = intention to treat analysis; Vit = vitamin; Zn = zinc; Cu = copper; Se = selenium; Ala = alanine; Arg = arginine; Asp = asparaginic acid; Cys = cysteine; Phe = phenylalanine; Glut = glutamine; Gly = glycine; His = histidine; Iso = isoleucine; Meth = methylalanine; Prol = proline; Ser = serine; Thre = threonine; Tyr = tyrosine; Val = valine; Try = Trypsin; LTM = lean tissue mass; ALM = appendicular lean mass; CSA = cross-sectional area; SMI = skeletal muscle index; EPA = eicosapentaenoic acid; DHA = docosahexaenoic acid; MRI = magnetic resonance imaging; ACSA = appendicular cross-sectional area; LITW = light-intensity training with whey protein supplementation; HRTW = heavy resistance training with whey protein supplementation; qCSa = quadriceps cross-sectional area.

## Data Availability

The data presented in this study are available on request from the corresponding author.

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
