# Peer review of "Increasing Muscle Mass in Elders through Diet and Exercise: A Literature Review of Recent RCTs"

_foods, 2023, doi:10.3390/foods12061218_

Round 1
Reviewer 1 Report
I read the review “
Increase Muscle Mass in Elders through Diet and Exercise: a literature review” with a lot of interest and attention and here are my comments/evaluation to the editor and authors.
This topic is very important and there are many important gaps to be solved. However, there is a lot of information about this topic in the literature. Many clinical trials and meta-analysis about the effect of exercise training and dietary interventions on muscle mass of elders; thus it was not clear to me across the introduction what this manuscript is adding.
Because there is a lot of information and all the interventions access the muscle mass in the same way, I don`t see why the authors did not run a systematic review with meta-analysis to be more conclusive. The 3 tables presented are very difficult to interpret, and they look more like text in a table format; which do not facilitate either the subjective interpretation. Because there is no systematic review we do not know if all studies were included leading to considerable bias. I could name a number of RCT of my group that tested chronic exercise intervention effects of muscle mass of older adults that are not listed. Therefore, it looks like a biased selection of papers to present in a very subjective way (no data-analysis).
In many parts of the text, it was not clear why the information was there and how it was connected to the rest of the text. And clarity was missing in many parts of the text, for example when talking about a cut-off for something not described (L66-71). What is the cut-off for?
A few references were missing in important statements such as: “Maintaining or increasing muscle mass is a key component for improving daily activities as well as sports performance in activities of daily living and in sports.” Or “Risk of falls, loss of independence and premature death has been gradually rising in the last decades among elderly population.”
The correct references were not mentioned, such as reference 10 does not prove it:
“Ageing is associated with a generalized deterioration of physiological function, ac-96 companied by a progressive decline in skeletal muscle mass and strength which progressively leads to functional impairment, increased disability and dependency”
There was misuse of references that did not conclude what is in the sentence. “Increased muscle 90 mass results from the accumulation of small amounts of protein in response to each bout of exercise combined with nutrient intake [9]“
Some sentences were written with no clarity “Physical inactivity appears to interact 86 and influence the extent to which fiber-type transformations occur“
Author Response
This topic is very important and there are many important gaps to be solved. However, there is a lot of information about this topic in the literature. Many clinical trials and meta-analysis about the effect of exercise training and dietary interventions on muscle mass of elders; thus it was not clear to me across the introduction what this manuscript is adding.
Because there is a lot of information and all the interventions access the muscle mass in the same way, I don`t see why the authors did not run a systematic review with meta-analysis to be more conclusive. The 3 tables presented are very difficult to interpret, and they look more like text in a table format; which do not facilitate either the subjective interpretation. Because there is no systematic review we do not know if all studies were included leading to considerable bias. I could name a number of RCT of my group that tested chronic exercise intervention effects of muscle mass of older adults that are not listed. Therefore, it looks like a biased selection of papers to present in a very subjective way (no data-analysis).
A: Thank you very much for your precious comments. We have taken them under serious consideration in order to improve our manuscript. For that reason, we systematically searched the Pubmed online database by creating a search strategy to retrieve all relative randomized controlled trials. We defined specific included and excluded criteria as described in the methods and we selected articles from the last five years in order to summarize recent research data.
In many parts of the text, it was not clear why the information was there and how it was connected to the rest of the text. And clarity was missing in many parts of the text, for example when talking about a cut-off for something not described (L66-71). What is the cut-off for?
A: We rewrite many parts of the text trying to be clear. We were also more specific about the cut-off values (L.71-75)
A few references were missing in important statements such as: “Maintaining or increasing muscle mass is a key component for improving daily activities as well as sports performance in activities of daily living and in sports.” Or “Risk of falls, loss of independence and premature death has been gradually rising in the last decades among elderly population.”
A: We add all the necessary references in the text, especially in the examples you mentioned.
The correct references were not mentioned, such as reference 10 does not prove it: “Ageing is associated with a generalized deterioration of physiological function, ac-96 companied by a progressive decline in skeletal muscle mass and strength which progressively leads to functional impairment, increased disability and dependency”
A: We change the ref 10 with a reference that proves this state.
There was misuse of references that did not conclude what is in the sentence. “Increased muscle 90 mass results from the accumulation of small amounts of protein in response to each bout of exercise combined with nutrient intake [9]“
A: We change the ref 9 with a reference that proves this state.
Some sentences were written with no clarity “Physical inactivity appears to interact 86 and influence the extent to which fiber-type transformations occur“
A: We delete this sentence.

Reviewer 2 Report
This review paper illustrates the independent and combined effects of diet and exercise on skeletal muscle mass in the elderly. In its present form, the review concludes that although the results of the analysed articles are often controversial, probably, the combination of exercise and high protein intake can improve muscle mass in old people.
The paper is well written and clear.
The literature reported is appropriate, although partially incomplete.
We have two suggestions.
Firstly, a short explanation regarding the effect of dietary supplementation on muscle metabolism should be discussed.
Indeed, metabolism is the sum of all chemical reactions that occur in living organisms, including ingestion, digestion and transportation of substances into and between different cells. The chemical reactions of metabolism are organized into metabolic pathways, in which one specific chemical compound is transformed through a series of steps into another molecules. Each step is facilitated by a specific enzyme, (Proteins), Co-Enzymes (i.e.: Vitamins), Co-Factors (i.e.: magnesium, calcium, selenium, etc.) and Intermediates (i.e..: HMB) which work synergically. [Cellular metabolism and disease: what do metabolic outliers teach us? Cell. 2012 Mar 16; 148(6): 1132–1144. Lehninger. 1975.Biochemestry].
Nutritional compounds are molecules that work synergically with a specific effect which influence the reactions of metabolism and consequently functional and/or anatomical effects. Consequently, a dietary supplementation with a specific mixture of different molecules involved at different steps in the muscle metabolism according to human needs should be suggested in the conclusion as possible further research.
Secondly research has shown that exercise therapy and dietary supplementation with a specific mixture of amino acids have similar histopathological, biochemical an functional changes in elderly patients (BioMed Research International. Vol 2014, article ID 34163. http://dx.doi.org/10.1155/2014/341603).
We believe that this article should be cited and discussed in the paper.
Author Response
Firstly, a short explanation regarding the effect of dietary supplementation on muscle metabolism should be discussed.
Indeed, metabolism is the sum of all chemical reactions that occur in living organisms, including ingestion, digestion and transportation of substances into and between different cells. The chemical reactions of metabolism are organized into metabolic pathways, in which one specific chemical compound is transformed through a series of steps into another molecules. Each step is facilitated by a specific enzyme, (Proteins), Co-Enzymes (i.e.: Vitamins), Co-Factors (i.e.: magnesium, calcium, selenium, etc.) and Intermediates (i.e..: HMB) which work synergically. [Cellular metabolism and disease: what do metabolic outliers teach us? Cell. 2012 Mar 16; 148(6): 1132–1144. Lehninger. 1975.Biochemestry].
Nutritional compounds are molecules that work synergically with a specific effect which influence the reactions of metabolism and consequently functional and/or anatomical effects. Consequently, a dietary supplementation with a specific mixture of different molecules involved at different steps in the muscle metabolism according to human needs should be suggested in the conclusion as possible further research.
A: Thank you very much for your precious comments. We add to the text a part that discussed the supplementation on muscle metabolism in the introduction and also in the conclusions (3rd paragraph).
Secondly research has shown that exercise therapy and dietary supplementation with a specific mixture of amino acids have similar histopathological, biochemical an functional changes in elderly patients (BioMed Research International. Vol 2014, article ID 34163. http://dx.doi.org/10.1155/2014/341603).
We believe that this article should be cited and discussed in the paper.
A: We discussed this citation in the introduction.

Reviewer 3 Report
Comments:
1. What is the current proportion of older adults with low-mass muscle or related conditions such as sarcopenia due to low muscle mass? It is suggested that relevant literature be added to the preface to highlight the significance of the author's concerns.
2. Abbreviations need to be defined the first time they appear in the text, such as CT, FFM, MRI and so on.
3. Section “2. Low muscle mass among different / various populations”. Here, different groups of people should include infants, young people, and the elderly. The author only briefly introduces the situation of the elderly under pathological conditions, and the content is not full and rich.
4. Section “3. Increase muscle mass in the elderly”. Here, the title is not a good match for the content. "Increasing muscle mass" should include causes and methods in content, rather than focusing on the disease.
5. Section “5. Nutrition and Dietary Supplements”. The authors only describe the effects of proteins, amino acids, and vitamins on muscle mass. Do carbohydrates and lipids also have an effect on muscle mass? The author should add to the relevant discussion.
6. Reasonable exercise time is also very important for the muscle mass of the elderly, and it is recommended to increase the relevant discussion.
7. The authors discuss the effects of diet and exercise on muscle mass in older adults. The diet focuses on protein and amino acids, it is recommended that this information be reflected in the title. For example, "Increase Muscle Mass in Elders through High-Protein Diet and Exercise" may be more appropriate.
8. The prevalence of sarcopenia in older people of different ages is different, such as the prevalence of 60 years of age and age over 80 years is very different. The authors should add recommendations for diet or exercise for older adults of different ages.
Author Response
- What is the current proportion of older adults with low-mass muscle or related conditions such as sarcopenia due to low muscle mass? It is suggested that relevant literature be added to the preface to highlight the significance of the author's concerns.
A: We add a meta-analysis about the prevalence of sarcopenia in the introduction.
- Abbreviations need to be defined the first time they appear in the text, such as CT, FFM, MRI and so on.
A: We defined the abbreviations which appeared the first time in the text.
- Section “2. Low muscle mass among different/various populations”. Here, different groups of people should include infants, young people, and the elderly. The author only briefly introduces the situation of the elderly under pathological conditions, and the content is not full and rich.
A: We delete this section as it was not necessary for the manuscript to discuss other population groups and we transfer the content in the introduction.
- Section “3. Increase muscle mass in the elderly”. Here, the title is not a good match for the content. "Increasing muscle mass" should include causes and methods in content, rather than focusing on the disease.
A: We delete this section as the aim of this section was not clear and we discussed the causes of the decrease of muscle mass and the methods for the increase of muscle mass in the rest of the manuscript.
- Section “5. Nutrition and Dietary Supplements”. The authors only describe the effects of proteins, amino acids, and vitamins on muscle mass. Do carbohydrates and lipids also have an effect on muscle mass? The author should add to the relevant discussion.
Α: We change the content and we discussed the effect of other macro- and micro-nutrients based on the included studies.
- Reasonable exercise time is also very important for the muscle mass of the elderly, and it is recommended to increase the relevant discussion.
Α: We discussed about the importance of the time as well as the other characteristics of the exercise in the discussion and the conclusions.
- The authors discuss the effects of diet and exercise on muscle mass in older adults. The diet focuses on protein and amino acids, it is recommended that this information be reflected in the title. For example, "Increase Muscle Mass in Elders through High-Protein Diet and Exercise" may be more appropriate.
Α: We change the discussion about nutrition. So, we do not focus only on protein and amino-acids, but we also discussed about MCTs and vitamins. As a result, “High-Protein Diet” will not be in accordance in the content of the review.
- The prevalence of sarcopenia in older people of different ages is different, such as the prevalence of 60 years of age and age over 80 years is very different. The authors should add recommendations for diet or exercise for older adults of different ages.
Α: There are no different recommendations established for older adults and elders. We mentioned that in the text.

Reviewer 4 Report
Dear Authors,
The paper is a literature review. It is not a systematic review.
Therefore, no evaluation was made according to the inclusion and exclusion criteria. Therefore, the desired studies were included and continued. Studies from different study types and time intervals are included in the tables. At the same time, the fact that individuals have different ages and diseases in the studies presented in the tables is also a matter of bias.
It is a literature review, but the boundaries of this literature review were not defined.
You should add a limitation section.
It has some typos. You should check for typos.
Author Response
Dear Authors,
The paper is a literature review. It is not a systematic review.
Therefore, no evaluation was made according to the inclusion and exclusion criteria. Therefore, the desired studies were included and continued. Studies from different study types and time intervals are included in the tables. At the same time, the fact that individuals have different ages and diseases in the studies presented in the tables is also a matter of bias.
It is a literature review, but the boundaries of this literature review were not defined.
You should add a limitation section.
It has some typos. You should check for typos.
Αnswer to reviewer:
Thank you very much for your precious comments. We have taken them under serious consideration in order to improve our manuscript. For that reason, we systematically searched the Pubmed online database by creating a search strategy to retrieve all relative randomized controlled trials. We defined specific included and excluded criteria as described in the methods and we selected articles from the last five years in order to summarize recent research data.
We also checked and correct the typos.

Round 2
Reviewer 1 Report
Dear authors and editor, unfortunately I can`t recommend publication of this manuscript due to a few issues that I previously described and were not solved and are considerable impactful for the quality of the manuscript and therefore the message it will pass. The main limitation for me, is the lack of precision in the information reported that was often followed by poor references and make us loose the trust in the authors affirmations. I felt the authors were many times stretching the truth and because it is a discursive review that authors are just summarizing the papers they choose to summarize, it is very important to be correctly assertive.
I could not find how the paper brought any novelty or solved any of the important gaps that remain to be solved within this topic.
Please see below my replies (R):
This topic is very important and there are many important gaps to be solved. However, there is a lot of information about this topic in the literature. Many clinical trials and meta-analysis about the effect of exercise training and dietary interventions on muscle mass of elders; thus it was not clear to me across the introduction what this manuscript is adding.
Because there is a lot of information and all the interventions access the muscle mass in the same way, I don`t see why the authors did not run a systematic review with meta-analysis to be more conclusive. The 3 tables presented are very difficult to interpret, and they look more like text in a table format; which do not facilitate either the subjective interpretation. Because there is no systematic review we do not know if all studies were included leading to considerable bias. I could name a number of RCT of my group that tested chronic exercise intervention effects of muscle mass of older adults that are not listed. Therefore, it looks like a biased selection of papers to present in a very subjective way (no data-analysis).
A: Thank you very much for your precious comments. We have taken them under serious consideration in order to improve our manuscript. For that reason, we systematically searched the Pubmed online database by creating a search strategy to retrieve all relative randomized controlled trials. We defined specific included and excluded criteria as described in the methods and we selected articles from the last five years in order to summarize recent research data.
R: Considering the main outcome was assessed by BIA, DXA, MRI, or CT, and the technology of last 5 years are exactly the same of many years ago, there is no reason to give more value of the papers of last five years and ignore the older ones. It is just confirming an important source of bias in this review.
In many parts of the text, it was not clear why the information was there and how it was connected to the rest of the text. And clarity was missing in many parts of the text, for example when talking about a cut-off for something not described (L66-71). What is the cut-off for?
A: We rewrite many parts of the text trying to be clear. We were also more specific about the cut-off values (L.71-75)
R: There was no description of cut-off on these lines.
A few references were missing in important statements such as: “Maintaining or increasing muscle mass is a key component for improving daily activities as well as sports performance in activities of daily living and in sports.” Or “Risk of falls, loss of independence and premature death has been gradually rising in the last decades among elderly population.”
A: We add all the necessary references in the text, especially in the examples you mentioned.
R: The references for the first sentence (“Maintaining or increasing muscle mass is a key component for improving daily activities as well as sports performance in activities of daily living and in sports.”) do not evidence what is in the sentence. Ref 15 only prove that resistance training increase muscle strength and hypertrophy with different loads and ref 16 do not show direct evidence for improving daily activities or sports performance with increase in muscle mass and strength. Also, I mentioned these sentences as examples of an issue that happened a few times and I did not see other changes marked in the text.
The correct references were not mentioned, such as reference 10 does not prove it: “Ageing is associated with a generalized deterioration of physiological function, ac-96 companied by a progressive decline in skeletal muscle mass and strength which progressively leads to functional impairment, increased disability and dependency”
A: We change the ref 10 with a reference that proves this state.
R: Excellent, reference 1 is now appropriate for the statement.
There was misuse of references that did not conclude what is in the sentence. “Increased muscle 90 mass results from the accumulation of small amounts of protein in response to each bout of exercise combined with nutrient intake [9]“
A: We change the ref 9 with a reference that proves this state.
R: good.
Some sentences were written with no clarity “Physical inactivity appears to interact 86 and influence the extent to which fiber-type transformations occur“
A: We delete this sentence.
R: ok.
Author Response
Dear authors and editor, unfortunately I can`t recommend publication of this manuscript due to a few issues that I previously described and were not solved and are considerable impactful for the quality of the manuscript and therefore the message it will pass. The main limitation for me, is the lack of precision in the information reported that was often followed by poor references and make us loose the trust in the authors affirmations. I felt the authors were many times stretching the truth and because it is a discursive review that authors are just summarizing the papers they choose to summarize, it is very important to be correctly assertive.
I could not find how the paper brought any novelty or solved any of the important gaps that remain to be solved within this topic.
A: Dear reviewer,
Thank you very much for your comments. However, we were really sorry that you were not convinced by the changes we have made in our manuscript and you think that there is a bias in the content. In order to avoid bias, we designed a search strategy for retrieving articles to provide a systematic way to retrieve published studies, and also defined inclusion and exclusion criteria to ensure homogeneity of included studies and avoid selection bias. As this is a review of the current literature, we tried to summarize the data obtained from randomized clinical trials; RCTs are the ideal type of study to include for evidence-based medicine. In addition, there is to date no published review (systematic or not) that summarizes the current data on muscle mass gain in older adults through exercise, diet and/or in combination. Thus, this leads us to believe that our manuscript is novel and useful. Finally, we add in the title that our manuscript is a review of recent RCTs to avoid misleading.
Please see below my replies (R):
This topic is very important and there are many important gaps to be solved. However, there is a lot of information about this topic in the literature. Many clinical trials and meta-analysis about the effect of exercise training and dietary interventions on muscle mass of elders; thus it was not clear to me across the introduction what this manuscript is adding.
Because there is a lot of information and all the interventions access the muscle mass in the same way, I don`t see why the authors did not run a systematic review with meta-analysis to be more conclusive. The 3 tables presented are very difficult to interpret, and they look more like text in a table format; which do not facilitate either the subjective interpretation. Because there is no systematic review we do not know if all studies were included leading to considerable bias. I could name a number of RCT of my group that tested chronic exercise intervention effects of muscle mass of older adults that are not listed. Therefore, it looks like a biased selection of papers to present in a very subjective way (no data-analysis).
A: Thank you very much for your precious comments. We have taken them under serious consideration in order to improve our manuscript. For that reason, we systematically searched the Pubmed online database by creating a search strategy to retrieve all relative randomized controlled trials. We defined specific included and excluded criteria as described in the methods and we selected articles from the last five years in order to summarize recent research data.
R: Considering the main outcome was assessed by BIA, DXA, MRI, or CT, and the technology of last 5 years are exactly the same of many years ago, there is no reason to give more value of the papers of last five years and ignore the older ones. It is just confirming an important source of bias in this review.
A: The evaluation tools were not the criterion that made us decide to include articles from the last five years. We specify the inclusion criteria for the studies. The criterion to include studies from the last five years is common to similar studies due to large number of published studies. There is a huge literature on the topic, which is really difficult to retrieve all these articles and summarize them all. Already the number of integrated studies is large and covers a wide range of aspects in regards to nutrition and/or exercise.
In many parts of the text, it was not clear why the information was there and how it was connected to the rest of the text. And clarity was missing in many parts of the text, for example when talking about a cut-off for something not described (L66-71). What is the cut-off for?
A: We rewrite many parts of the text trying to be clear. We were also more specific about the cut-off values (L.71-75)
R: There was no description of cut-off on these lines.
A: We find the points where we referred cut-off values in our manuscript and we made sure that provided the correct cut-off values. We couldn’t find cut-off values in the lines you initially referred.
A few references were missing in important statements such as: “Maintaining or increasing muscle mass is a key component for improving daily activities as well as sports performance in activities of daily living and in sports.” Or “Risk of falls, loss of independence and premature death has been gradually rising in the last decades among elderly population.”
A: We add all the necessary references in the text, especially in the examples you mentioned.
R: The references for the first sentence (“Maintaining or increasing muscle mass is a key component for improving daily activities as well as sports performance in activities of daily living and in sports.”) do not evidence what is in the sentence. Ref 15 only prove that resistance training increase muscle strength and hypertrophy with different loads and ref 16 do not show direct evidence for improving daily activities or sports performance with increase in muscle mass and strength. Also, I mentioned these sentences as examples of an issue that happened a few times and I did not see other changes marked in the text.
A:The changes have been made. We changed the reference to the sentence "Maintaining or increasing muscle mass is a key component for improving daily activities as well as athletic performance in activities of daily living and sports, with one more appropriate than the others. We have checked the references and find no other missing or abusive references in the manuscript.

Reviewer 3 Report
This paper has been well revised and greatly improved and could be accepted for publication. This manuscript is in general clearly written, but still need to revise in order to avoid careless typos/mistakes in technical concepts, language expressions, formats and styles.
Author Response
Reviewer 3
This paper has been well revised and greatly improved and could be accepted for publication. This manuscript is in general clearly written, but still need to revise in order to avoid careless typos/mistakes in technical concepts, language expressions, formats and styles.
Dear Reviewer,
Thank you very much for your comments. We check our manuscript for typos/mistakes in technical concepts, language expressions, formats and styles and we made the necessary changes in the manuscript with track changes.
